# Current Status of Oligonucleotide-Based Protein Degraders

**DOI:** 10.3390/pharmaceutics15030765

**Published:** 2023-02-24

**Authors:** Po-Chang Shih, Miyako Naganuma, Yosuke Demizu, Mikihiko Naito

**Affiliations:** 1Graduate School of Pharmaceutical Sciences, The University of Tokyo, Tokyo 113-0033, Japan; 2National Institute of Health Sciences, Kawasaki 210-9501, Japan; 3Graduate School of Medical Life Science, Yokohama City University, Yokohama 230-0045, Japan

**Keywords:** degrader therapeutics, targeted protein degradation, PROTAC, SNIPER, nucleic acids, proteases, chimera

## Abstract

Transcription factors (TFs) and RNA-binding proteins (RBPs) have long been considered undruggable, mainly because they lack ligand-binding sites and are equipped with flat and narrow protein surfaces. Protein-specific oligonucleotides have been harnessed to target these proteins with some satisfactory preclinical results. The emerging proteolysis-targeting chimera (PROTAC) technology is no exception, utilizing protein-specific oligonucleotides as warheads to target TFs and RBPs. In addition, proteolysis by proteases is another type of protein degradation. In this review article, we discuss the current status of oligonucleotide-based protein degraders that are dependent either on the ubiquitin–proteasome system or a protease, providing a reference for the future development of degraders.

## 1. Introduction

Protein-targeting strategies have been widely employed for drug development in the last two decades, and occupancy-driven mechanisms of action (MoAs) are the main means of inhibiting or enhancing the functions of proteins. The occupancy-driven MoA represents an inhibition or enhancement strategy that works by fitting a ligand to a binding pocket on the protein [1]. Traditional targeted therapy relies on occupancy-driven MoAs to achieve specified goals for disease treatment. The ligands used for occupancy-driven mechanisms can be small molecules or antibodies, as well as oligonucleotides, and the resulting binding affinities between the ligands and protein targets are optimized to be sufficiently potent for the induction of satisfactory pharmacological effects.

Oligonucleotide-binding proteins such as transcription factors (TFs) and RNA-binding proteins (RBPs) have long been considered undruggable targets [2,3]. This is because one of the requirements for drugs with occupancy-driven MoAs is that there is a strong binding potency between the drugs and the proteins [1]. Generally, TFs exhibit shallow and flat protein surfaces that do not facilitate strong intermolecular interactions, and they do not have ligand-binding sites. Additionally, TFs naturally interact with downstream promoter sequences with dissociation constants at low nanomolar ranges [4]. These hurdles mean that small molecule binders are unable to bind to TFs effectively. Although antibody drugs can exhibit strong interactions with TFs and RBPs at nanomolar ranges if properly designed, they have long been known to be inappropriate for targeting, mainly because their structures are overly large and complex to pass through the cell membrane using the current techniques [5]. Harnessing oligonucleotides has therefore been identified as a promising solution for targeting such difficult proteins, since they can demonstrate potent inter-molecular interactions at nanomolar ranges by efficient delivery into cells using, for example, transfection or transduction.

Targeted protein degradation (TPD) technologies that use the ubiquitin–proteasome system (UPS), represented by proteolysis-targeting chimeras (PROTACs) [1,6], have evolved rapidly, and some degrader molecules have entered into clinical trials [7]. The UPS is an intrinsic protein clearance mechanism involving E1, E2, and E3 ligases that are widely present in cells. PROTAC molecules are bifunctional molecules containing a warhead binding to a protein of interest (POI), an E3 ligase recruiter, and a linker conjugated between the POI warhead and E3 recruiter. The resulting chimeric molecule is therefore able to bind to both the POI and the E3 ligase, leading to the close proximity of both macromolecules to transfer ubiquitin markers for the following proteasomal degradation (Figure 1). At present, only a few E3 ligases have been successfully harnessed to recruit disease-causing POIs, even though more than 600 of them have been identified [1,6]. Another reported class of TPD technology is reliant on proteases, which catalytically destroy POIs. Using proteases as a degradation machinery, the bifunctional molecules usually do not require a linking moiety, and in this respect differ from PROTAC-based technologies.

Recently emerging PROTAC technologies have been studied in the context of degrading cytoplasmic, nuclear, and membrane-associated proteins such as BCR-Abl, BRD4, and the receptor tyrosine kinases EGFR and FLT3 [8,9,10,11,12,13,14], and most of them utilize small molecules as warheads to recruit POIs. In addition, these technologies utilize oligonucleotides to design protein degraders for targeting TFs and other oligonucleotide-binding proteins, although this field is much less focused than that of small molecule-warheaded protein degraders. The oligonucleotides utilized can be single-stranded or double-stranded decoys, aptamers, RNA, or G-quadruplexes. In this review, we focus on the current understanding of the protein degraders that are usually designed to have oligonucleotides as warheads to bind to the POI, and which are generally referred to as oligonucleotide-warheaded PROTACs. In addition, more recent protease-mediated proteolysis technology is discussed.

## 2. PROATC-Based Protein Degraders Warheaded with Single-Stranded or Double-Stranded Decoys: TRAFTAC, OligoTRAFTAC, O’PROTAC, TF-PROTAC, and SNIPER

Cereblon (CRBN), von-Hippel-Lindau (VHL), and inhibitors of apoptosis proteins (IAPs) are E3 ligases that are commonly used in the field of protein degrader development [1,15]. The resulting chimeric protein degraders are collectively referred to as PROTACs or PROTAC molecules, although IAP-based PROTACs are sometimes called SNIPERs. TRAFTAC, OligoTRAFTAC, O’PROTAC, and TF-PROTAC are PROTAC-based technologies reported to induce transcription factor degradation; they were developed by different research groups using either VHL or CRBN as the E3 ligase, and these technologies use decoy oligonucleotides as their POI recruiters. Table 1 summarizes the currently disclosed oligonucleotide-based protein degraders.

The TRAFTAC developed by Samarasinghe et al. utilized a dCas9-HT7 fusion protein as an intermediary machinery (Figure 2). In this design for degradation molecules, a POI-specific double-stranded DNA (dsDNA) sequence is covalently linked to a dCas9-specific RNA sequence (abbreviated as dsDNA-RNA), while a VHL-E3 ligase binder is linked to an HT7 binder to generate a HaloPROTAC molecule. By co-incubating the dsDNA-RNA chimeric molecule with HaloPROTAC, the POI and VHL-E3 ligase are recruited in close proximity to complete ubiquitin transfer, thus inducing downstream proteasomal degradation. The TRAFTAC successfully degraded cancer-related NF-ĸB and brachyury TFs via VHL-dependent UPS in cervical cancer HeLa cells by NF-ĸB-TRAFTAC and Brachyury-TRAFTAC, respectively. To generate double-stranded DNA for recruiting POIs, a NF-ĸB binding sequence 5′-GGGAATTTCC-3′ and a brachyury binding sequence 5′-AATTTCACACCT-3′ were referenced. Since brachyury is essential for vertebrate notochord formation at early stages of embryonic development [27], the authors used zebrafish as an animal model in the in vivo studies, finding that the microinjection of Brachyury-TRAFTAC into zebrafish embryos at the single-cell stage induced tail formation defects [16].

OligoTRAFTAC is the second generation of TRAFTAC developed by the same group. OligoTRAFTAC used the same PROTAC design rationale as shown in Figure 1. One oligoTRAFTAC contains a POI-specific oligonucleotide, a VHL-E3 ligase binder, and a linker conjugated in between (Figure 2). In this oligoTRAFTAC, an alkyne–azide click reaction (a synthetic strategy that can decrease the laboriousness of oligoPROTAC synthesis) was adopted. The oligoTRAFTAC successfully degraded cancer-related cMYC and brachyury TFs by VHL-dependent UPS in HeLa and HEK293T cells, the degradation of which was abrogated by the E1 inhibitor MLN4924. To generate double-stranded warheads, a cMYC binding consensus sequence 5′-CACGTGGTTGCCACGTG-3′ and a brachyury binding DNA sequence 5′-AATTTCACACCTAGGTGTGAAATT-3′ were referenced. With its significantly improved handiness, the author group considered oligoTRAFTAC as a generalizable platform for other TFs, although washout experiments showed that the withdrawal of oligoTRAFTAC did not prolong degradation responses [17]. Again, the authors used zebrafish as an animal model in the in vivo studies, finding that microinjection of oligoTRAFTAC OT17 induced tail deformation in 70% of the injected zebrafish embryos [17]. Notably, the in vivo studies confirmed that phosphodiester bonds in the oligoTRAFTAC molecules are vulnerable in living animals due to nucleases including exonucleases and endonucleases, whereas phosphorothioate backbones in the oligoTRAFTAC molecules largely avoided nuclease attacks, showing nuclease stability.

At around the same time, Shao et al. reported a programmable oligonucleotide PROTAC which was named O’PROTAC (Figure 2). Whereas VHL was exclusively used in TRAFTAC and OligoTRAFTAC, these authors demonstrated that both VHL and CRBN are capable of inducing TFs degradation, and their degraders successfully degraded lymphoid enhancer binding factor 1 (LEF1) and ETS-related gene (ERG), the two oncogenic TFs [18]. Similarly to other oligonucleotide-warheaded PROTAC technologies, the programmability of O’PROTAC relies on known POI-specific DNA oligomers, which can be found on free databases [28,29], rendering their syntheses straightforward and efficient. LEF1 O’PROTAC molecules were designed using 18-mer double-stranded oligonucleotide referenced from 5′-TACAAAGATCAAAGGGTT-3′, in which the underscored bases are the LEF1 binding moiety and three additional nucleotides were added for the protection of oligo degradation by exonucleases-mediated hydrolysis [28,29]. On the other hand, ERG O’PROTAC molecules were designed using the 19-mer double-stranded oligonucleotide referenced from 5′-ACGGACCGGAAATCCGGTT-3′, in which the underscored bases are the ERG binding moiety and three additional nucleotides were added for the protection of oligo degradation. The LEF1 O’PROTAC molecule LEF1 OP-V1 was found to be the most effective in degrading the POI in prostate cancer PC-1 and DU145 cells. LEF1 OP-V1 inhibited prostate cancer cell proliferation in vitro and tumor growth in vivo using PC-1 and DU145 xenografted mice. The most effective ERG O’PROTAC molecule ERG OP-C-N1 degraded ERG protein and inhibited cancer cell growth in vitro in prostate cancer VCaP cell line [18].

Using a similar development rationale to that of oligoTRAFTAC, TF-PROTAC technology (Figure 2) was developed by Liu et al. To facilitate synthesis, the authors harnessed VHL as the E3 ligase and used an alkyne–azide click reaction to yield the TF-PROTAC molecules dE2F and dNF-ĸB. A single-stranded DNA sequence of 5′-TGGGGACTTTCCAGTTTCTGGAAAGTCCCCA-3′ was used as a warhead to target the subunit of NF-ĸB, p65, while double-stranded DNA 15-mers (the sense chain was 5′-CTAGATTTCCCGCG-3′ and the antisense chain was 5′-CTAGCGCGGAAAT-3′) were selected to bait cancer-related E2F. Since the NF-ĸB-specific single-stranded oligonucleotide forms a double-stranded hairpin structure through intra-dimerization, it confers resistance to exonuclease degradation in cells. Incubated in HeLa cells, the TF-PROTAC molecules dE2F and dNF-ĸB successfully degraded E2F and the subunit of NF-ĸB, p65, respectively, which were rescued by pre-incubation with the proteasome inhibitor MG132 and the VHL ligand VH032, as well as the depletion of the endogenous VHL E3 ligase. In addition, 24–72 h treatment with dE2F #16 and #17 at 25 μg/mL or dNF-ĸB #15 and #16 at 10 μg/mL demonstrated markedly anti-proliferative and anti-tumorigenic effects in HeLa cells [19]. Notably, despite the success of TF-PROTAC in cancer cells, it remains unclear whether unwanted effects in normal cells (e.g., cytotoxicity) could be induced by the TF-PROTAC molecules. Moreover, it must still be determined whether the above-mentioned pharmacological effects can also be observed in living animals.

In addition to the use of VHL and CRBN, another family of E3 ligases known as IAPs has also been harnessed in the development of PROTACs [30,31,32]. Our research team synthesized a series of VHL-, CRBN-, and IAP-based PROTACs targeting estrogen receptor alpha (ERα) (Figure 2), a TF able to form transcriptional complexes on the promoter region to initiate gene expression [33]. Double-stranded DNA 21-mers were used as a decoy with sequences of 5′-GTCAGGTCACAGTGACCTGAT-3′ and 3′-CAGTCCAGTGTCACTGGACTA-5′. Following evaluation with Western blot, we found that only IAP-based PROTACs with a PEG3 linker, such as LCL-ER(dec), successfully destroyed the ERα in breast cancer MCF7 cells, while VHL- and CRBN-based ones did not induce degradation up to 10 μM. Our findings also showed that these SNIPER-type decoy degraders did not affect the protein levels of other transcription factors such as androgen receptor (AR), aryl hydrocarbon receptor (AhR), and NF-κB p65 [20]. The LCL-ER(dec)-induced reduction in ERα levels was abrogated by co-treatment with the proteasome inhibitor MG132 and the ubiquitin-activating inhibitor MLN7243. Intriguingly, our research team not only demonstrated that ERα can be degraded by oligonucleotide-warheaded molecules but also by peptide- and small-molecule-warheaded SNIPER molecules [8,34,35], confirming that one TF can be degraded by at least three kinds of warheads.

## 3. Aptamer-PROTAC Conjugates and Aptamer-Warheaded PROTAC

Aptamers are single-stranded oligonucleotides that are discovered using an in vitro method of SELEX (systematic evolution of ligands by exponential enrichment), and which usually have a length shorter than 100 nucleotides. Similarly to antibodies, they show high affinity and specificity against their targets by folding into unique three-dimensional conformations [36,37]; they have some advantages that antibodies do not have, for instance, low unwanted immunogenicity and toxicity, easy chemical synthesis and modification, rapid tissue penetration, and excellent stability [38,39]. The aptamer-related immunogenicity is lower than that of antibodies, perhaps because aptamers generally contain sugars modified at their 2′-positions; therefore, Toll-like receptor-mediated innate immune responses are abrogated [40,41]. According to the current understanding obtained from clinical studies, aptamer-related adverse events are rare [39]. The SELEX technique constructs large libraries of degenerate oligonucleotides, which are iteratively and alternately partitioned for protein target binding [41]. Inspired by antibody-drug conjugates, aptamer-PROTAC conjugates (APCs) are a group of chimeric molecules that link an aptamer with a conventional PROTAC molecule (Figure 2 and Figure 3). He et al. reported the first type of APCs linking an aptamer to the hydroxyl group of VHL ligands in a VHL-based PROTAC molecule. The hydroxyl group is critical to achieving effective binding of the VHL ligand to VHL-E3 ligase; therefore, the authors designed a cleavable linkage method to conjugate the aptamer by using a disulfide bond and a carbon anhydride group. The endogenous glutathione cleaves the disulfide bond, and the resulting mercapto functional group subsequently attacks the electrophilic carbon anhydride group to release the conventional PROTAC (Figure 3) [21]. In the study, a 26-base, guanine-rich, single-stranded DNA aptamer, AS1411, which specifically binds to nucleolin, a cell membrane protein that is widely expressed in tumor cells, was utilized. Hence, the use of AS1411 is believed to have an add-on benefit that shows selectivity for tumor cells over normal cells [42]. The sequence of AS1411 is 5′-GGTGGTGGTGGTTGTGGTGGTGGTGG-3′. Moreover, AS1411 itself demonstrates anti-proliferative activities against breast cancer, glioma, renal cell carcinoma, and leukemia [43,44,45,46], further affirming the value of using this agent for anti-cancer purposes. The authors selected BET family proteins for proof-of-concept studies. The BET family proteins BRD2, 3, and 4 are key epigenetic regulators that recognize and bind to acetylated histones, causing the conversion of chromatin into a conformation ready for transcriptional elongation via RNA polymerase II. BET family proteins are therefore regarded as promising antitumor targets because of their functions in the regulation of gene transcription and their mediation of the transcription of associated genes [47]. APR is the BET-targeting APC developed by the group, and it was found to be able to enter into MCF7 cells via micropinocytosis, which is the main mechanism whereby cells internalize the macromolecule AS1411 [42]. MCF7 cells overexpress nucleolin and were considered nucleolin-positive in the studies. In vitro studies showed that BRD2, 3, and 4 were successfully degraded by the APR in MCF7 cells, and BRD4 was affected the most strongly among the BET family proteins [21]. The cell selectivity of APR for cancer cells over normal cells was demonstrated by 50% growth inhibitory concentrations (GI_50_) of 59.8 nM in MCF7 cells versus 3.13 μM in normal MCF-10A cells. In vivo distribution studies using MCF7 xenografted models showed that intravenous injection of APR led to drug accumulation, mainly in the liver, kidneys, and tumor sites. The tumor growth inhibition rate of APR was determined to be around 77%, much more potent than AS1411, and APR did not cause any weight loss or other observable side effects [21].

Recently, AS1411 was also adopted by Zhang et al., who conjugated it with a VHL ligand to make a PROTAC molecule named ZL216, synthesized using a DBCO–azide click reaction. ZL216 successfully induced nucleolin degradation in MCF7 and BT474 breast cancer cells but not in MCF10A cells, and it displayed excellent serum stability with a half-life of 70.5 h, as well as water solubility. In addition, ZL216 and AS1411 showed consistently low nanomolar dissociation constants against MCF7 and BT474 cells, implying that ZL216 has an excellent ability to bind to breast cancer cells. Tail injections of ZL216 in MCF7- and BT474-xenografted mice showed that it possessed the same ability to target tumors as AS1411, and it downregulated nucleolin, which AS1411 could not achieve. The use of a VHL binder, (S, R, S)-AHPC-PEG3-azide, as a competitor against ZL216 significantly rescued the effect of ZL216-induced nucleolin degradation in MCF7 and BT474 cells. Additionally, ZL216 inhibited the proliferation and migration of breast cancer cells in vitro [22]. This study further validates the selectivity of AS1411 for cancer cells over normal cells, and aptamers can be harnessed as POI recruiters in addition to small molecule binders.

More recently, Chen et al. reported a light-controllable, AS1411-warheaded PROTAC molecule, the opto-dNCL#T1, for use against breast cancer cells (Figure 2) [23]. As shown in Figure 2, the opto-dNCL#T1 contains a quencher-conjugated photolabile oligonucleotide (Q-CP) and a fluorescent dye Cy3-conjugated, AS1411-warheaded PROTAC molecule dNCL#T1 (dNCL#T1-Cy3). Both Q-CP and dNCL#T1-Cy3 are single-stranded and complement each other to form double-stranded opto-dNCL#T1; upon light activation by UVA irradiation for 10 min, Q-CP detaches for the release of dNCL#T1. Unlike Zhang et al., who used VHL as the E3 ligase, Chen et al. used CRBN. Consistently with the degradation effect of dNCL#T1 on nucleolin, the findings showed that opto-dNCL#T1 degraded nucleolin upon UVA irradiation. Both light-activated opto-dNCL#T1 and dNCL#T1 demonstrated similar cytotoxicity and anti-migration effects in MCF7 and MDA-MB-231 cells at 24–72 h. Moreover, this study developed another class of aptamer-warheaded PROTAC molecule dDNMT1#0 to degrade DNA methyltransferase 1 (DNMT1) using a DNMT1-specific aptamer [48]. The dDNMT1#0 treatment significantly inhibited the proliferation of HeLa and HL-60 cells. Overall, this study confirms that this type of PROTAC methodology is a feasible strategy for degrading proteins other than nucleolin, and that a photolabile component can be conjugated to yield a light-controllable, aptamer-warheaded PROTAC molecule.

## 4. RNA-PROTACs: RNA-Warheaded PROTACs for Targeting RNA-Binding Proteins

A single cell is known to contain over 1500 RNA-binding proteins (RBPs) that are evolutionally conserved, and defects in them causes many diseases, including cancer [49,50,51,52]. Like TFs, these proteins have also been difficult to target with the use of conventional small-molecule inhibitors [53]. Ghidini et al. designed RNA-based protein degraders by conjugating small RNA mimics, a linker, and a peptide sequence derived from HIF-1α, to report the first PROTAC molecules for use against RBPs. HIF-1α is reported to be recruited by the VHL ligase [54], thereby forming a close proximity to the E3 ligase. One class of synthesized RNA-PROTAC molecules was warheaded with chemically modified 5′-AGGAGAU-3′ sequences, in which 5′-AGGAGAU-3′ is a conserved sequence present in microRNAs to which stem cell factor Lin28 is bound. The loss-of-function variant of LIN28 has been found in Parkinson’s disease (PD) patients [55], making it a protein target for PD. Additionally, to further validate the proof of concept, the authors developed another class of RNA-warheaded PROTAC molecules based on 5′-UGCAUGU-3′ to target splicing factor RBFOX1. Abnormalities in RBFOX1, which is also known as A2BP1, have been associated with mental disorders [56]. The respective RNA-warheaded PROTAC molecules successfully destroyed not only LIN28 but also RBFOX1 in myelogenous leukemia K562 and embryonic kidney-derived HEK293 cells [24]. This study provides an in vitro proof-of-concept reference for using RNA as a bait to destroy RBPs, paving the way to target these relatively new protein targets. However, certain shortcomings, such as the instability of RNA oligomers and the requirement of RNA secondary structures for proper interactions with RBPs, should be carefully considered.

## 5. G4-PROTAC: G-Quadruplex-Warheaded Protein Degraders

G-quadruplexes, also known as G4, are atypical nucleic acids with four-stranded secondary structures stabilized by cyclical Hoogsteen hydrogen bonding and central cationic coordination; they have been implicated in regulating gene transcription, replication, repair, and other processes [57,58,59,60]. Patil et al. reported their G4-warheaded PROTAC molecules using either a VHL ligand (VH032) or a CRBN ligand (pomalidomide) to target the DEAH-box helicase RHAU, a type of G4-binding protein also known as DHX36 and G4R1 [61,62]. Studies have shown that RHAU binds preferentially to all-parallel-stranded G4 with high affinity [63,64], and T95-2T (sequence TT(GGGT)_4_) is an all-parallel-stranded G4 [65]. With the use of T95-2T and either VH032 or pomalidomide, the resulting G4 PROTACs (named G4-V and G4-P, respectively) developed by the authors were able to induce RHAU degradation in HeLa cells. G4-P was more potent than G4-V under the same sequence and linker length, and 24 h incubation was the optimal time point for the maximal induction of RHAU degradation. Similarly, an alkyne–azide click reaction was adopted to synthesize G4 PROTACs [25]. This study provides an in vitro proof-of-concept reference for using G4 as a warhead to degrade G4-binding proteins; therefore, TFs that bind to G4 are potentially degradable if G4-based PROTACs are carefully selected.

## 6. Apt-Clean: Bispecific Aptamers Act as Degraders for Targeting Extracellular Receptors

Previously discussed examples of oligonucleotide-based protein degraders (Figure 2) require the laborious and tedious workload of conjugating an oligonucleotide, a linker, and an E3 binder to generate a protein degrader. Apt-clean, reported by Hoshiyama et al., is another class of protein degradation technology; here, a bispecific aptamer is designed to recruit both POI and a protease into close proximity, thereby leading to the degradation of POI by the protease (Figure 4) [26]. Apt-clean does not generally need a linker. Thrombin, a multifunctional serine protease that regulates pro-coagulant and anti-coagulant functions [66], was selected as a protease for Apt-clean proof-of-concept studies. This protease shows a degree of promiscuity in the recognition of multiple macromolecular substrates [67,68], and the substrate specificity of thrombin involves two electropositive surfaces on the protease: the fibrinogen-recognizing and heparin-binding exosites [69,70]. A variety of thrombin-specific aptamers have been studied, from which HD1 and HD22 aptamer sequences were used in the Apt-clean research [71,72]. HD1 and HD22 are, respectively, bound to exosites 1 and 2 of thrombin, where exosite 1 is a binding site for various thrombin substrates such as fibrinogen, factor V, and factor VIII, while exosite 2 is a binding site for heparin and facilitates interactions with antithrombin III and heparin cofactor II [67,68]. Notably, the binding of HD1 on exosite 1 and HD22 on exosite 2 does not hinder the protease catalytic site, leaving the catalytic activity of thrombin unaffected; accordingly, these two aptamers could be ideal thrombin recruiters when developing Apt-clean bispecific aptamers. The sequence of HD1 is 5′-GGTTGGTGTGGTTGG-3′, while that of HD22 is 5′-AGTCCGTGGTAGGGCAGGTTGGGGTGACT-3′. A 38-mer DNA aptamer, SL38.2, was used to specifically target fibroblast growth factor receptor 1 (FGFR1, the POI of the research) with nanomolar affinity [73]. The sequence of SL38.2 is 5′-CGATCGATGGATGGTAGCTCGGTCGGGGTGGGTGGGTTGGCAATCGATCG-3′. By conjugating SL38.2 with HD1 or HD22, Apt-clean bispecific aptamers 1 to 4, i.e., SL38.2-HD1, SL38.2-HD22, HD1-SL38.2, and HD22-SL38.2, were synthesized. In a cell-free environment incubated with thrombin and FGFR1-Fc fusion protein, the findings indicated that Apt-clean 2 and 4, in which HD22 is respectively tethered to the 3′ or 5′ terminal of SL38.2, demonstrated the strongest degradation efficiency among the tested samples. FGFR1-expressing 3T3-L1 cells were used for further validation. With the co-incubation of Apt-clean 2 with thrombin in 3T3-L1 cells, FGFR1 phosphorylation by its ligand FGF2 was abrogated. In addition, this study synthesized a 10-dT linker named Apt-clean 5 (SL38.2-10-dT-HD22) and tethered it in between SL38.2 and HD22; however, the linker did not yield stronger degradation efficiency compared to Apt-clean 2 [26]. Overall, this study opens up a new direction for using bispecific oligonucleotides for TPD drug development.

## 7. Perspectives

Drug development using intrinsic protein degradation machineries has evolved rapidly in the recent years. Our perspectives, provided below, discuss the recent progress of TPD, focusing specifically on those with oligonucleotide-based protein degraders.

At present, two types of protein degradation machinery, proteasome and thrombin, have been adopted to develop oligonucleotide-based degraders, PROTACs and Apt-cleans, respectively. They are compensatory in terms of the localization of target proteins because proteasomes degrade proteins intracellularly, while thrombin targets extracellular proteins and proteins on the cell surface. The nature of oligonucleotides, which have difficulty in penetrating cells, seems to lend itself well to Apt-clean. However, the bispecific nature of Apt-clean can be easily applied to PROTACs once the issue of cellular uptake is overcome.

For PROTAC-based degrader molecules, the success of the protein degraders does not only depend on proper degrader design; the abundance of the E3 ligase also plays an important role. This review article addresses the cell types and E3 ligases these authors used in their biological evaluations, providing a cell selection reference for evaluating new PROTAC-based degrader molecules. In addition, Békés et al. has comprehensively reviewed E3 family members that show tissue and cell-type specificity as well as tumor enrichment [15]. Some E3 ligases are enriched in muscle tissues, such as KLHL40 and KLHL41, while CRBN and VHL are ubiquitous in the body. Another report from Cowan and Ciulli focuses on E3 ligase substrate specificity [74]. Accordingly, one should carefully select E3 ligases and appropriate cells when developing and evaluating protein degraders. Their review articles can be referred to for more detailed information.

It has been reported that one TF is able to recognize several similar oligonucleotide sequences, and one oligonucleotide sequence could be shared among several TFs [27]. For example, TCF3 recognizes an E-box consensus sequence similar to that of c-Myc [17]. This strongly indicates that avoiding off-target effects requires a well-designed oligonucleotide sequence for one TF. Multiple large-scale studies have provided conclusive strategies for selecting specific oligonucleotide sequences [75,76,77,78]; however, the conclusions are considered incomplete [27]. Accordingly, one could conduct a comprehensive reference search and utilize online databases such as EPD (The Eukaryotic Promoter Database) to increase the likelihood of using highly specific oligonucleotides [79].

Although the studies discussed above did not carry out systematic investigations into the effects of the length of oligonucleotides on the degradation efficiency and off-target effects, the length of oligonucleotides could also play a role. For example, the propensity for self-annealing of single-stranded DNA increases depending on its length, which may result in decreased degradation efficiency due to its inappropriate secondary structure. On the contrary, shorter oligonucleotides may cause off-target effects due to the high probability of overlapping nucleotide sequences among other TFs. It remains unclear whether the length of oligonucleotides affects degradation efficiency and induces off-target effects; this, as well as the question of how many nucleotides is considered sufficient, must be systematically studied in the future.

Undoubtedly, some researchers may be concerned about degradation of oligonucleotides by DNases within the body, hence, reducing the potentiality of such a class of protein degraders. In this regard, a few strategies have been explored to enhance oligonucleotide stability: for instance, the chemical modification of inter-nucleoside linkages, the chemical modification of deoxyriboses, the chemical modification of nucleobases, and derivatization at the 5′ or 3′ ends [80,81,82,83]. Additionally, peptide nucleic acids (PNA) could also be used as a complimentary strand so as to decrease DNase attacks [84]. The main difference between PNAs and natural nucleic acids lies in the backbone structure, where PNA displays 2-aminoethyl glycine linkages in place of natural phosphodiester bonds of DNA/RNA [84,85]. Intriguingly, PNA can tolerate attacks from not only DNases but also peptidases/proteases [85]. It should be noted that, after using non-natural nucleotides, biosafety and binding potency should be carefully (re-)examined [86,87,88], before moving forward to clinical trial settings.

Most small-molecule-warheaded PROTAC-based degraders focus on anti-cancer effects; this is also the case for oligonucleotide-warheaded degraders, as discussed above. Although TFs represent a major class of essential proteins that maintain the proliferation and tumorigenesis of cancer cells, as revealed by the cancer dependency map project (DepMap) [89], other disease areas that are also PROTAC-applicable include neurodegenerative disorders. Alzheimer’s disease has been considered a promising therapeutic area, since its pathology is believed to be caused by tau-protein-induced neurofibrillary tangles in the brain. The removal of the tau protein is recommended as an attractive target to degrade, and a few PROTAC molecules have been synthesized, showing effective pre-clinically degradation capabilities for tau [90]. Disease areas beyond cancer and neurodegenerative disorders have received much less attention, suggesting that PROTAC-based technologies need to be extended to other treatment applications.

Precision medicine is an approach to achieving tailored treatment or diagnosis for an individual. The first evidence of precision medicine originated thousands of years ago [91], and targeted therapy for cancer diseases is a well-known example that falls within this category [92]. However, even with the development of a diversity of therapy modalities in the last half century, we have yet to develop a strategy that achieves satisfactory treatment outcomes for cancer diseases due to side effects. It is known that drug side effects are attributed to unwanted cytotoxicity to normal cells; hence, precisely attacking cancer cells treated with anti-cancer therapies while leaving normal cells unaffected is considered critical when developing anti-neoplastic precision medicines. To achieve such precision targeting, oligonucleotide-based PROTAC molecules could be a feasible treatment method. The PROTAC methodology requires the selection of a protein related to oncogenesis, which makes this methodology a type of targeted therapy. This constitutes the first layer of precision targeting. The second layer of precision targeting can be fulfilled by using a cancer-cell-specific E3 ligase. Multiple studies have focused on this specified research field, because commonly used VHL and CRBN E3 ligases are generally found in all tissues [93,94,95]. In terms of oligonucleotide warheads, double- and single-stranded oligonucleotides, as well as aptamers, can be used for PROTACs. However, as evidenced previously, breast-cancer-cell-specific aptamer AS1411 can be conjugated with a PROTAC molecule (as inspired by antibody–drug conjugates) as a delivery means to selectively target cancer cells over normal cells [21]; this constitutes the third layer of precision targeting. One could use structurally modified AS1411 as an alternative [96,97,98,99,100,101,102], because AS1411 was withdrawn from clinical trials [103]. The main reason for the withdrawal of AS1411 from clinical trials was its insufficient anti-cancer activity; meanwhile, its safety issue was found to be good [44]. The fourth layer is attributable to a photolabile component conjugated to PROTAC molecules. Light can be used to activate photolabile PROTAC molecules at target tumor sites only, so as to leave unspecified sites unaffected by medicines. Taken together, four layers of precision targeting are identified that, in theory, could lead to more satisfactory treatment outcomes for precision medicine.

Oligonucleotides are not only used as warheads for recruiting POIs in PROTAC-based degraders, but are also likely to act as linkers. Most linkers consist of polymers (e.g., polymethylene, polyethylene glycol, or amide) to conjugate ligands for POIs and E3 ligases [104,105]. The disadvantages of using polymers as linkers are as follows: (1) Controlling the degree of polymerization is a laborious and time-consuming process, since researchers are required to construct linker libraries; (2) the high number of rotatable bonds in linkers means that the spacing distance cannot be precisely tuned; (3) the high number of rotatable bonds in linkers also leads to the indecisive directionality of PROTAC molecules. These factors have slowed the progress of PROTAC compound screening. In contrast, using oligonucleotides such as DNA as linkers for generating PROTACs may not have the abovementioned downsides. For example, DNA can be made through a solid-phase DNA synthesizer that can generate a variety of DNA sequences cost-effectively and within a short timeframe [106,107,108,109,110,111]. In addition, DNA consists of four types of repeated deoxynucleotides connected through the same phosphate diester bond; hence, the length of the DNA polymer can be precisely controlled with the spacing distance of two adjacent deoxynucleotides at around 0.33 nm [112]. Moreover, short DNA sequences usually take a helix form in nature; therefore, their directionality is predictable. Zhao et al. reported their DNA-modularized strategy to precisely control the length and orientation of bivalent molecules [112]. In their report, one muscarinic acetylcholine receptor (MR), M1R, was used for in vitro proof-of-concept studies. The activation of the M1Rs by xanomeline lessens cognitive impairments and negative symptoms in Alzheimer’s disease and schizophrenia; however, it yields severe side effects as it simultaneously activates other subtypes of MRs [113,114,115,116,117]. Therefore, these authors constructed bivalent compounds consisting of xanomeline (an agonist in the orthosteric ligand-binding site of M1Rs), benzyl quinolone carboxylic acid (BQCA, a modulator in the allosteric ligand-binding site of M1Rs), and short DNA sequences as linkers in between. In doing so, M1Rs are selectively activated but other subtypes of MRs are not, especially M4Rs, because they are the dominant MRs subtype activated by xanomeline. Two bases (a combination of A, T, C, and G) were used in the linker, and the authors found that BQCA-A-A-xanomeline was the most selective compound for activating M1Rs; 72 h incubation in serum or DNase 1 did not markedly degrade the compound, indicating it could retain plasma instability in vivo [112]. Although the examples reported by Zhao et al. are not PROTAC-related molecules or protein degraders, their strategy is believed to be applicable to designing linkers for protein degraders. It could be argued that using oligonucleotide sequences as linkers in PROTAC molecules may generate trivalent compounds if the sequences are specific to one kind of TF. In other words, to avoid off-target effects in TFs, the oligonucleotide sequences must be carefully designed. The use of oligonucleotide sequences as linkers in PROTAC drug development warrants validation.

Despite the promising outlook for PROTAC-based degrader molecules, they face drug metabolism and pharmacokinetics (DMPK) issues, even with some small-molecule-warheaded PROTACs being evaluated in early phase clinical trials, as our understanding of the physicochemical properties of compounds beyond “Rule of Five” (Ro5) space remains limited. PROTAC-based degrader molecules generally have the physicochemical properties of high molecular weight, high polar surface area, and a high number of rotatable bonds compared to Ro5 space, and these features lead to poor solubility and permeability [118]. The poor solubility in aqueous solution and larger size in nature cause difficulties in measuring the acid dissociation constant (pK_a_) and distribution coefficient (logD) if the methods commonly used to evaluate small molecules are adopted. This has led to the reporting of alternative methodologies, such as the determination of pKa of fragments of the entire degrader and ChromlogD, as a replacement for logD [119]. In computational prediction campaigns, PROTAC-based degrader molecules are considered new chemical entities that are not well set up for training to yield reliable prediction results for absorption, distribution, metabolism, elimination (ADME), and toxicity. To obtain ADME results in in vitro and in vivo settings, alternative approaches have been suggested for PROTAC-based molecules in place of the methods conventionally used for small molecules, yet optimization work may also be needed [119]. Overall, compared to small molecule entities, the prediction and testing of DMPK properties for protein degraders require more research efforts to overcome development bottlenecks [120].

Unlike small molecules, the development of oligonucleotides for therapeutic use requires more effort, as the number of approved small molecule drugs outweighs the number of oligonucleotide-based drugs [121,122]. Oligonucleotide-warheaded PROTAC-based molecules fall within this category. Due to their large size and negatively charged nature, oligonucleotides cannot cross the negatively charged cell membrane as easily as small molecules can. Additionally, achieving efficient delivery to target sites other than the liver is a major difficulty. These issues, in addition to the DMPK issues, are believed to constitute further hurdles to clinically developing oligonucleotide-warheaded PROTAC-based molecules. At present, chemical modification, bioconjugation, and the use of nanocarriers are the primary strategies aiming to allow oligonucleotides to penetrate the cell membrane for clinical use [122]. Usually, in preclinical settings, transfection reagents are used to internalize oligonucleotides into cells in in vitro assays, and most of the oligonucleotide-warheaded PROTAC-based degrader molecules discussed above used reagents that are not developed for clinical use. Therefore, one should carefully evaluate the clinical potential of developing oligonucleotide-warheaded PROTAC-based molecules. Another aspect that needs to be considered is whether the oligonucleotides used could trigger the cyclic GMP–AMP synthase-stimulator of interferon genes pathway, for short, the cGAS-STING pathway, which is an initiator of the immune response of human bodies when fighting against foreign DNA or RNA [123]. The studies discussed above did not investigate this area.

To date, no oligonucleotide-warheaded PROTAC-based degrader molecules have been evaluated in clinical settings; meanwhile, the first small-molecule-warheaded PROTAC ARV-110 entered into clinical evaluations in 2019 [15]. With this review article summarizing our current understanding of oligonucleotide-based protein degraders, as well as some challenges together with possible solutions, the development progress of the relevant degrader molecules could be accelerated.

## Figures and Tables

**Figure 1 pharmaceutics-15-00765-f001:**
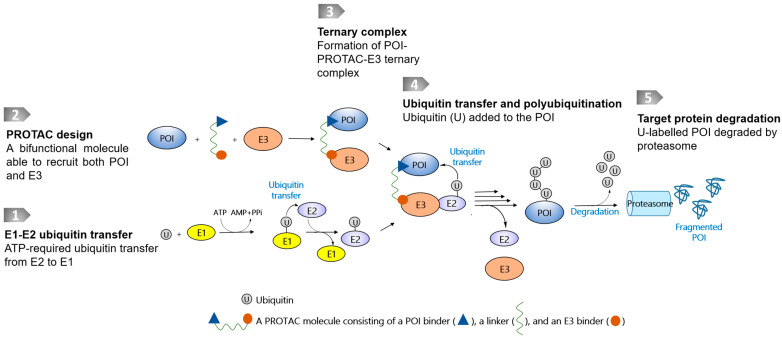
Schematic diagram of a PROTAC molecule and its degradation pathway. The degradation pathway is reliant on E1, E2, and E3 ligases and proteasome. The E3s are recruited to POI by PROTACs, thereby inducing ubiquitylation and proteasomal degradation.

**Figure 2 pharmaceutics-15-00765-f002:**
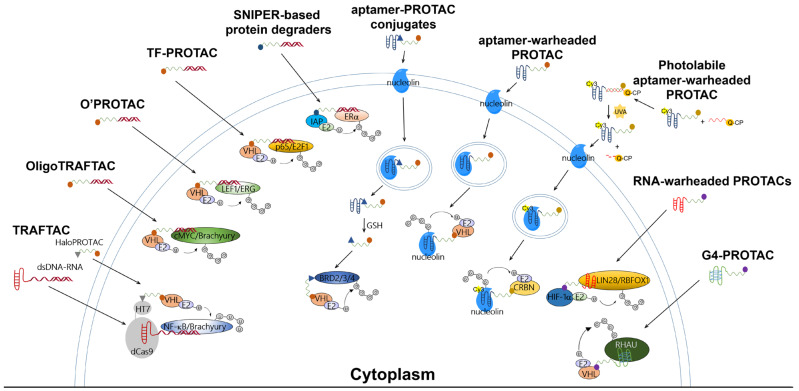
Ubiquitination mechanisms of reported PROTAC-based technologies for targeting TFs or RBPs. These PROTAC molecules essentially consist of an E3 binder, a linking moiety, and a POI binder. The degradation of the POI is UPS-dependent.

**Figure 3 pharmaceutics-15-00765-f003:**
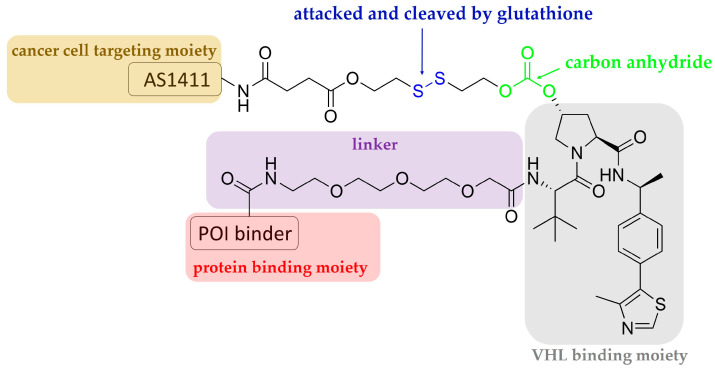
Schematic representation of the design of an aptamer–PROTAC conjugate.

**Figure 4 pharmaceutics-15-00765-f004:**
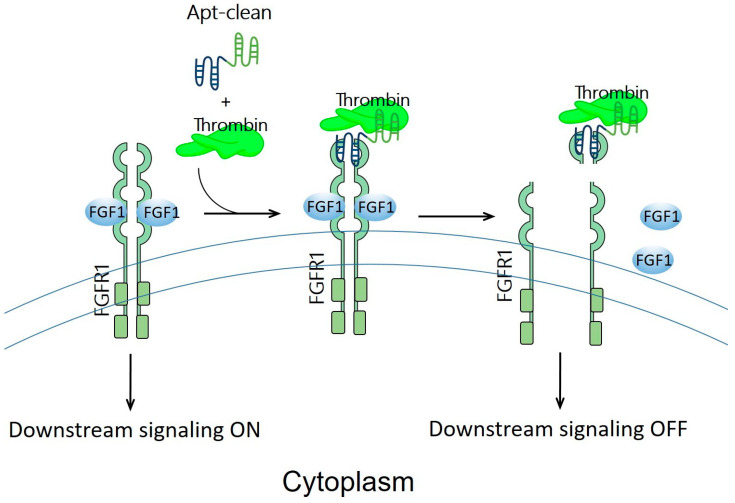
Schematic representation of the design of Apt-clean bispecific aptamer molecules. Essentially, Apt-clean bispecific aptamers consist of an aptamer for a POI (e.g., FGFR1) colored in blue, and another aptamer for a protease (e.g., thrombin) colored in green; generally, a linking moiety is not needed in between the aptamers. The degradation pathway is not UPS-dependent; instead, it relies on the protease activity of thrombin to initiate catalytic proteolysis.

**Table 1 pharmaceutics-15-00765-t001:** Currently disclosed oligonucleotide-based degraders with the E3 ligase used, target proteins, cell lines tested, and core DNA/RNA sequences used. The DNA/RNA sequences may be extended or shortened, or the bases may be chemically modified. More information can be found in the corresponding references.

Names	E3 Ligases	POIs	Cell Lines	Ref.
	Core DNA/RNA Sequences Used			
TRAFTAC	VHL	NF-ĸB and brachyury	HeLa	[16]
	To generate double-stranded DNA for recruiting POIs, a NF-ĸB binding sequence 5′-GGGAATTTCC-3′ and a brachyury binding sequence 5′-AATTTCACACCT-3′ were referenced.			
OligoTRAFTAC	VHL	cMYC and brachyury	HeLa and HEK293T	[17]
	To generate double-stranded warheads, a cMYC binding consensus sequence 5′-CACGTGGTTGCCACGTG-3′ and a brachyury binding DNA sequence 5′-AATTTCACACCTAGGTGTGAAATT-3′ were referenced.			
O’PROTAC	VHL and CRBN	binding factor 1 (LEF1) and ETS-related gene (ERG)	PC-3	[18]
	A 18-mer oligonucleotide, 5′-TACAAAGATCAAAGGGTT-3′, was referenced for generating double-stranded DNA specific for targeting LEF1.A 19-mer oligonucleotide, 5′-ACGGACCGGAAATCCGGTT-3′, was referenced for generating double-stranded DNA specific for targeting ERG.			
TF-PROTAC	VHL	E2F and the subunit of NF-ĸB, p65	HeLa	[19]
	A single-stranded DNA sequence of 5′-TGGGGACTTTCCAGTTTCTGGAAAGTCCCCA-3′ was used as a warhead to target the subunit of NF-ĸB, p65, while double-stranded DNA 15-mers (sense chain was 5′-CTAGATTTCCCGCG-3′ and the antisense chain was 5′-CTAGCGCGGAAAT-3′) were selected to bait cancer-related E2F.			
SNIPER	IAP	ERα	MCF7	[20]
	The 21-mer, ERα-specific, double-stranded DNA was referenced from 5′-GTCAGGTCACAGTGACCTGAT-3′.			
Aptamer-PROTAC conjugates	VHL	BRD2, 3, and 4	MCF7	[21]
	The 26-mer AS1411 was 5′-GGTGGTGGTGGTTGTGGTGGTGGTGG-3′.			
Aptamer-warheaded PROTAC	VHL	nucleolin	MCF7 and BT474	[22]
	The same AS1411 sequence as above was used.			
Light controllable, aptamer-warheaded PROTAC	CRBN	nucleolin	MCF7 and MDA-MB-231	[23]
	The same AS1411 sequence as above was used.			
RNA-warheaded PROTAC	VHL/HIF-1α	LIN28 and RBFOX1	K562	[24]
	5′-AGGAGAU-3′ was for Lin28, while 5′-UGCAUGU-3′ was for RBFOX1.			
G4-PROTAC	VHL and CRBN	RHAU	HeLa	[25]
	RHAU-targeting G4 sequence was TT(GGGT)_4_.			
Apt-clean	No E3 used; instead, thrombin protease	FGFR1	3T3-L1	[26]
	The aptamer sequence of HD1 was 5′-GGTTGGTGTGGTTGG-3′, while the aptamer sequence of HD22 was 5′-AGTCCGTGGTAGGGCAGGTTGGGGTGACT-3′.A 38-mer DNA aptamer of FGFR1-binding SL38.2 was 5′-CGATCGATGGATGGTAGCTCGGTCGGGGTGGGTGGGTTGGCAATCGATCG-3′.			

The underlines in the DNA sequences represent core binding moieties specific for corresponding protein targets.

## Data Availability

Not applicable.

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
