# Peer review of "Current Status of Oligonucleotide-Based Protein Degraders"

_pharmaceutics, 2023, doi:10.3390/pharmaceutics15030765_

Round 1

Reviewer 1 Report

Shih et al., highlight the different methods that were used to degrade undruggable proteins (especially transcription factors) using oligonucleotides. 

This review article has a good structure and communicates the science to the larger community in a comprehensible language.

The selection of articles that were highlighted in the review is informative and recent. The citations are highly relevant. Table 1 discusses the different PROTACs made with oligos that are useful. 

I thoroughly enjoyed reading this work, especially about the aptamer PROTACs.

I have some minor suggestions to address.

  1. In Figure 1, there is a lot of white space. I would alter the structure of the figure and even recommend them to make appealing figures (same holds for Figure 3). Kindly make the figure more informative.
  2. Though Figure 2 is highly informative, the fonts within the figure are pixelated. Please use some professional illustrator software to generate a high-quality image.
  3. Is there any chance for the oligos to trigger the immune response via the cGAS STING pathway? Please comment about the safety of the approach (Justify why you 'mentioned' that oligos have low immunogenicity over antibodies)
  4. Please discuss whether the size of the DNA influences the degradation efficacy and off-target effects (if any).
  5. Is there any scope to use PNAs to solve the stability issue with DNases?
  6. Is there any advantage in degrading proteins via Proteases over Proteasomes? Please speculate.
  7. In what ways oligonucleotide based PROTACs better than siRNAs?

This article is highly relevant to the targeted protein degradation field. I highly recommend this manuscript for publication with minor changes. Congratulations to the team.

Reviewer 2 Report

The manuscript by Shih and colleagues is a review that covers the topic of oligonucleotide-based protein degraders. This is an interesting field that continues to grow and should find a broad audience.

 The writing is problematic and already the first few lines of the abstract do not form a complete sentence and many other instances throughout the manuscript need to be carefully edited and corrected (e.g. even the title to section 2.2 “PROATC-Based…”; “destruct” should likely be “destroy” throughout most of the manuscript).

In figure 2, the authors should increase the font size of the smaller labels so that the reader can see these easily without having to zoom in. Otherwise, this is a nice overview of the technologies. Typos like “TF-RPOTAC”, also in the text describing this figure in the main text, need to be corrected.  Statements like “TRAFTAC is regarded as a chemical tool.” should be explained in more detail or deleted.  

Page 5, lines 164-165: The authors should provide more insight to such statements. “superior anti-proliferative effect in cells” – what kind of cells? Only cancer cell lines? Were normal, healthy cells tested to check for potential unwanted activities of the TF-PROTACs? Can these be specifically delivered to not harm healthy cells? These are all important information that would help guide the reader through this field and identify strengths and weaknesses of such approaches.

Page 6, line 188: fewer than 100 what? Nucleotides?

Section 3: Here, the authors explain the chemical reactions used to generate the VHL-based PROTAC molecule. A figure illustrating these reactions would be very helpful and greatly improve the understanding of these reactions.

What is the activity of AS1411 against healthy cells?

Section 6, line 311; what is “brinogen-recognition”? Fibrinogen? The authors should clarify and correct this.

In the perspectives, the authors write “carefully select appropriate cells when evaluating degraders”. This is an important point that suggests limitations of the method and should be further discussed and clarified as this may have strong impact on the clinical translation of such approaches.

Why was AS1411 withdrawn from clinical trials? This information should be included in the perspectives section.

The authors mention the challenges of clinical translation, and it would be useful for the reader if the authors could also mention some of the strategies that could be implemented to overcome these challenges.

Reviewer 3 Report

In this review article, Shih et al outline the design and applications of various oligonucleotide-based protein degraders. Building on the success of small molecule targeting PROTACS, this review expands this PROTAC design into DNA and RNA targeting moieties for difficult to target drug targets including transcription factors and RNA binding proteins. The review presents an emerging and interesting field of study. With a few minor revisions this review will be prepared for publication.

1.     In the introduction section, the terms PROTCAC and SNIPER are linked together as if they are separate technologies, but Table 1 indicates that SNIPER is a subcategory of PROTACS. If PROTAC and SNIPER are two different technologies, the authors need to more clearly define their differences and what differentiates a SNIPER from a PROTAC. If SNIPER is a subtype of PROTAC, then the introduction section should discuss PROTACS in general and then they can explain SNIPER as a subgroup with all of the other PROTACS varieties outlined in Table 1.

2.     Figures and Tables

a.     Figure 2: The writing on the individual protein, small molecule, and biomolecules are too small to read. This Figure has so much information and greatly assists with interpreting the written text about each PROTAC. Figure 2 should be made larger (entire page) to better accommodate the amount of information shown in this figure.

b.     Figure 3 and Figure 1B are nearly identical and do not present enough novel information to warrant both of these figures. Since there is only one example of a protease-based targeting system, Figure 1B seems unnecessary as there is no reason to generalize this technology but instead could just be replaced with Figure 3.

c.     The authors list the DNA/RNA sequences used in the targeting. This is valuable information to include but would be better formatted as a separate table pulling together all of the oligonucleotide sequences from throughout the article. Some accompanying information about key factors to consider in oligonucleotide design or maybe resources for oligonucleotide design would also be warranted.

3.     The perspective section needs an introductory paragraph to help organize the logic of this section. In this introductory paragraph, the authors need to lay out what are key things to consider when designing these molecules. Then they can use this outline as they go through each factor individually in the perspective section.

4.     The final conclusion paragraph paints a rosy picture about the future of oligonucleotide based PROTACs and future clinical applications. This reviewer thinks that this section is too optimistic given the preceding paragraph about the uncertainty and severe limitations of drug delivery for this oligonucleotide based therapeutics. If the authors wish to present this rosy view, they also need to more clearly explain how this delivery limitation will be overcome and which delivery techniques are most promising for these oligonucleotide based therapeutics.

Round 2

Reviewer 2 Report

The authors have addressed my concerns and should be congratulated on a much-improved manuscript.